

# The risk of hospitalization for respiratory tract infection (RTI) in children who are treated with high-dose IVIG in Kawasaki Disease: a nationwide population-based matched cohort study

Wei-Te Lei[1], Chien-Yu Lin[1,2], Yu-Hsuan Kao[3], Cheng-Hung Lee[4], Chao-Hsu Lin[1], Shyh-Dar Shyur[3], Kuender-Der Yang[3] and Jian-Han Chen[5]

[1] Department of Pediatrics, Hsinchu MacKay Memorial Hospital, Hsinchu, Taiwan
[2] College of Medicine and Veterinary Medicine, the University of Edinburgh, Scotland, UK
[3] Department of Pediatrics, MacKay Children's Hospital, Taipei, Taiwan
[4] Department of General Surgery, Buddhist Dalin Tzu Chi Hospital, Chia-Yi, Taiwan
[5] Department of General Surgery, E-Da Hospital, Kaohsiung, Taiwan

## ABSTRACT

**Background**. Kawasaki disease (KD) is an immune-mediated systemic vasculitis, and infection plays an important role in the pathophysiology of KD. The susceptibility to infectious disease in patients with KD remains largely unclear. This study aimed to investigate the risk of respiratory tract infection (RTI)-related hospitalizations in children with KD.

**Methods**. Data from the Taiwanese National Health Insurance Research Database was analyzed. We excluded patients with history of congenital abnormality, allergic diseases, or hospitalization history. Children with KD were selected as KD group and age- and sex-matched non-KD patients were selected as control group with 1:4 ratio. Both cohorts were tracked for one year to investigate the incidences of RTI-related hospitalizations. Cox regression hazard model was used to adjust for confounding factors and calculate the adjusted hazard ratio (aHR).

**Results**. Between January 1996 and December 2012, 4,973 patients with KD were identified as the KD group and 19,683 patients were enrolled as the control group. An obviously reduced risk of RTI-related hospitalizations was observed in KD patients (aHR: 0.75, 95% CI [0.66–0.85]). The decreased risk persisted through the first six-months follow-up period with a peak protection in 3–6 months (aHR: 0.49, 95% CI [0.37–0.64]).

**Conclusions**. KD patients had approximately half reduction of risk for RTI-related hospitalizations. The protective effects persisted for at least six months. Further studies are warranted to elucidate the entire mechanism and investigate the influences of intravenous immunoglobulin.

Corresponding author
Jian-Han Chen,
jamihan1981@gmail.com

# INTRODUCTION

Kawasaki disease (KD) is an important cause of systemic vasculitis and the main cause of heart disease in childhood (*Newburger et al., 2004*). Although the etiology is not well known, specific infection agents may cause genetic susceptibility in certain children (*Wang et al., 2005*; *Weng et al., 2017*). Globally, Taiwan has the third highest incidence of KD, second only to Korea and Japan (*Huang et al., 2009*). An aberrant immune response is also believed to be key in the preclinical and acute stages of this disease (*Hara et al., 2016*; *Matsubara, Ichiyama & Furukawa, 2005*). Treatment with high-dose intravenous immunoglobulin (IVIG) along with moderate to high doses of aspirin is the current gold standard (*Newburger et al., 2004*). In Taiwan, a high dose 2 g/kg/dose IVIG is the standard treatment of KD. While the mechanisms underlying the beneficial effects of IVIG remain unknown, its broad anti-inflammatory effect is believed to be an important factor. Coronary artery aneurysm is the main concern of KD but children with KD also have higher risk of some systemic diseases, such as atopic dermatitis (*Abrams et al., 2017*; *Wei et al., 2014*). Clinically, no obvious risks of infectious diseases are observed in patients with KD and children with KD are not regarded as immunocompromised patients. However, studies investigating the infectious risk are limited and the susceptibility to infectious disease in patients with KD remains largely unclear.

Human products of IVIG is made by the pooled human plasma of approximately a thousand blood donors (*Hemming, 2001*; *Perez et al., 2017*). Initially, it was intended to protect patients with primary immunodeficiency (PID) against infection. The ability of anti-inflammation and immune-modulation of IVIG has been recognized and its use is increasing in a variety of diseases (*Siberil et al., 2007*; *Wong & White, 2016*). The US Food and Drug Administration has approved the indications of IVIG use: treatment in patients with PID; prevention in patients with hypogammaglobulinemia and recurrent bacterial infection due to B-cell chronic lymphocytic leukemia; prevention of pneumonitis and graft-versus-host disease following bone marrow transplantation; reduction of severe infection in HIV-infected children; prevention of coronary artery aneurysms in KD; and decrease bleeding in idiopathic thrombocytopenic purpura (ITP) (*Perez et al., 2017*). IVIG is composed by "protective" immunoglobulin and the recommendation of IVIG use in treatment or prophylaxis in patients with various kinds of immunodeficiency is well documented (*Ammann et al., 1982*; *Hemming, 2001*; *Keller & Stiehm, 2000*; *Malik, Giacoia & West, 1991*; *Orange et al., 2010*). However, the protective effects of IVIG against infection in other patients remain largely unclear. Preterm babies have relatively lower level of immunoglobulins and are prone to infection. Studies investigating the prophylactic effects of IVIG in preterm patients demonstrated a 3% reduction in sepsis and 4% reduction in any serious infection. But IVIG use was not associated with clinical outcomes (*Ohlsson & Lacy, 2013*). Further studies are required to elucidate the protective effects of IVIG.

In our daily practice, we found that children with KD seemed to have less severe infection after treatment of KD but no related studies were found in the literature. KD is an immune-mediated disease, and patients with KD may have different susceptibility to infectious

disease compared with general population. Furthermore, KD is the major indication for IVIG use and the protective effects of IVIG against infection may contribute to the observed decreased risk (*Perez et al., 2017*). Therefore, a nationwide population-based study was conducted to investigate the incidences of hospitalization for respiratory tract infections (RTI) in children with KD and explore the potentially protective effects of IVIG.

## MATERIALS AND METHODS

### Data sources

Data were retrieved from the National Health Insurance Research Database (NHIRD) of the National Health Research Institutes. The Taiwan's National Health Insurance (NHI) is a nationwide program implemented in March 1995. Approximately 99% of 23.74 million residents in Taiwan have been enrolled in this program (*Liu et al., 2017*). The database contains universal files and longitudinal medical records from 1996 to 2013, such as demographic characteristics, inpatient and outpatient data, dates of admission, diagnostic codes, and medical prescriptions. All diseases are coded with the International Classification of Disease, Ninth Revision, Clinical Modification (ICD-9-CM) (*American Hospital Association et al., 1990*). This study was reviewed and approved by the Institutional Review Board of Buddhist Tzu Chi Hospital, Dalin, Taiwan (IRB approval number: B10503021). The institutional review board exempted consent requirement.

### Study design and population

Two cohorts involved in our study were chosen from the NHIRD: KD group and control group. Figure 1 describes the cohort study framework. We extracted patient data regarding inpatient expenditures by admissions from the NHIRD database between January 1, 1996 and December 31, 2013. In order to minimize the potential bias of selection, patients with history of congenital abnormality, allergic diseases, or hospitalization history were excluded in both cohorts. Patients with KD were selected as KD group and the exclusion criteria included (1) recurrent KD, (2) birth before 1996, and (3) age $\geq$ 6 years old. (4) with congenital abnormality and prenatal disease, (5) with history of asthma, atopic dermatitis and allergic rhinitis, (6) with respiratory tract infection history. Meanwhile, patients without KD were selected as control group and the exclusion criteria comprised (1) age $\geq$ 6 years old, (2) with any hospitalization history including respiratory tract infections, congenital abnormality and prenatal disease, and (3) with history of asthma, atopic dermatitis and allergic rhinitis. Patients were randomly selected from non-KD cohort by 1:4 propensity score match with parameters including age and sex with patients from KD cohort. We followed both cohorts for one year to investigate the incidences of RTI-related hospitalizations. In this cohort study, we identified 17,580 patients who were admitted with a diagnostic code of KD (ICD-9, 446.1). Patients who were born before January 1, 1996 ($n = 2{,}007$) and those who were admitted after December 31, 2012 ($n = 955$) were excluded. In addition, 2,497 patients with recurrent KD, 354 patients aged $\geq$ 6 years, 16 patients with undetermined sex, 2,977 patients with congenital abnormalities and prenatal disease, 190 patients with past history of asthma, atopic dermatitis and allergic rhinitis, and 3,611 patients with history of respiratory tract infections were also excluded. Finally, 4,973
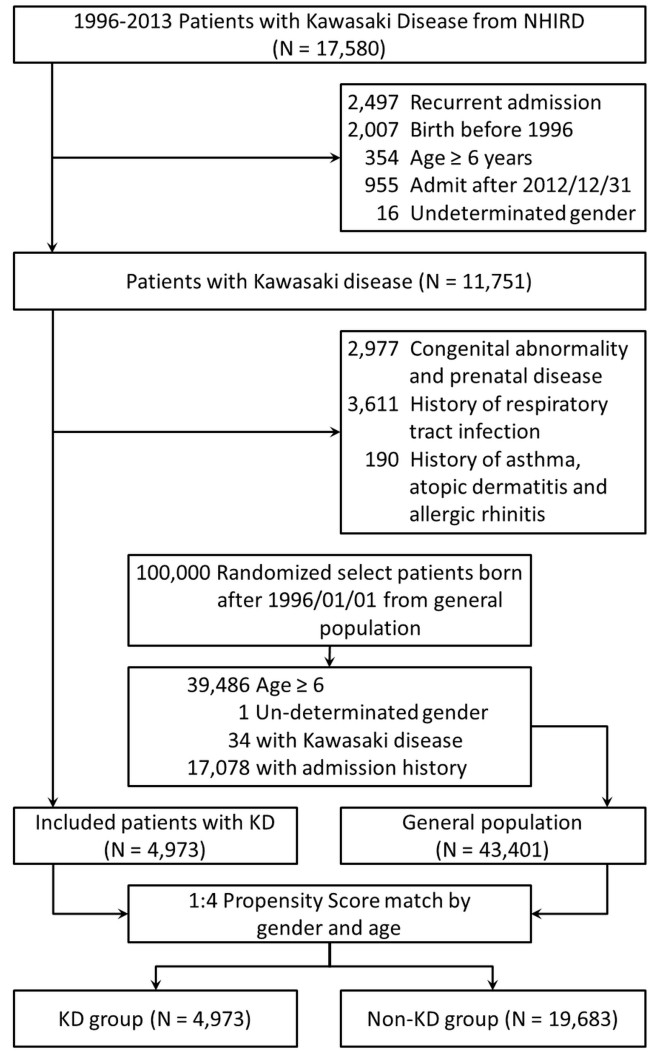

**Figure 1    The flow chart of enrollment of study participants.**

patients aged <6 years and newly diagnosed with KD were included. The uses of IVIG were validated by prescription records.

## Study outcomes and follow-up duration

All patients were followed up for one year after enrollment. Censoring was defined as withdrawal from the insurance program due to death, living abroad for >6 months, or missing appointments for >6 months. The primary outcomes were to determine admissions for RTI-related hospitalizations, including pneumonia, acute otitis media, acute bronchitis, influenza, and acute tonsillitis. To identify accompanied respiratory tract infection, associated comorbidities and congenital abnormalities, we searched diagnosis based on the categories of Clinical Classification Software codes (CCS) (Supplemental Information 1), which collapsed all ICD-9-CM's diagnosis and procedure codes into clinically meaningful categories that are useful for presenting descriptive statics (*Thompson*

*et al., 2006*). The congenital abnormality (CCs-Multiple-Diagnosis 14.x.x), prenatal disease (CCs-Multiple-Diagnosis 15.2–6), AOM (CCs-Multiple-Diagnosis 6.8.1), pneumonia (CCs-Multiple-Diagnosis 8.1.1), tonsillitis (CCs-Multiple-Diagnosis 8.1.3), bronchitis (CCs-Multiple-Diagnosis 8.1.4), and asthma were identified. (CCs-Multiple-Diagnosis 8.3.x). Other comorbidities including atopic dermatitis (ICD-9-CM codes 691.8) and allergic rhinitis (ICD-9-CM codes 477.9) were also identified. The reason for choosing pneumonia, AOM, bronchitis, and tonsillitis as the representation for RTI in this study is based on the top ten diagnoses of children requiring emergency care and subsequent hospitalization from a 10-year population-based nationwide analysis in Taiwan (*Jeng et al., 2014*).

## Statistical analysis

The incidence density rate of KD (per 1,000 person-years) was measured based on national live birth data. Differences in demographic characteristics, co-variables, and admissions for respiratory infections between IVIG and control groups were analyzed using categorical variables, the Student's $t$ test for continuous variables with normal contribution, and the Mann–Whitney $U$ test for continuous variables without normal contribution. The Kaplan–Meier methods with the log-rank test were used to compare the survival distributions between the cohorts. $P$ values of $<0.2$ were inserted into Cox proportional hazards analysis to quantify the risk of subsequent RTIs after adjusting for comorbidities. All statistical analyses were performed with SPSS (Statistical Package for Social Science) statistical software version 22.0 (IBM, Armonk, NY, USA). A two-sided $P$ value of $<0.05$ was considered statistically significant.

## RESULTS

We identified 4,973 patients as KD group and 19,683 patients as non-KD group between 1996 and 2013 in Taiwan. Table 1 demonstrated the baseline characteristics for the two cohorts. Both cohorts showed a similar distribution of age; 73.80% and 73.88% patients were aged <2 years in the KD and non-KD cohorts, and 26.20% and 26.12% patients were aged two and six years, in the KD and non-KD cohorts, respectively. The KD cohort had a similar distribution of gender with the non-KD cohort.

The cumulative incidence of all RTI-related hospitalizations was significantly lower in the KD cohort ($p < 0.001$, log-rank test) than in the non-KD cohort (Fig. 2). Furthermore, the cumulative incidence of hospitalizations due to pneumonia, AOM, and acute bronchiolitis were all lower in the KD cohort ($p = 0.010$, $p = 0.010$, $p = 0.033$, respectively) than in the non-KD cohort (Figs. 3–5). There was no significant difference in the cumulative incidence of hospitalization due to tonsillitis between both group ($p = 0.819$).

The hazard ratios of RTI-related hospitalizations were listed in Table 2. The overall incidence ratio of all RTI-related hospitalizations was 0.87-fold lower in the KD cohort (16.72 vs 19.18 per 1,000 person-months, 95% CI [0.76–0.99]; $p < 0.05$). Moreover, the incidence ratio of pneumonia-related hospitalizations was also lower in the KD cohort (15.79 vs 20.52 per 1,000 person-months, 95% CI [0.61–0.96]; $p < 0.05$). After adjusting for potential confounders, adjusted hazard ratios (aHRs) were lower in the KD cohort

**Table 1  Demographics between Kawasaki disease group and non-Kawasaki disease group.**

| Characteristics | KD (N = 4,973) | | Non-KD (N = 19,683) | | P value |
|---|---|---|---|---|---|
| | No. | % | No. | % | |
| **Age group** | | | | | 0.914 |
| 0–2 y | 3,670 | 73.80 | 14,542 | 73.88 | |
| 2–6 y | 1,303 | 26.20 | 5,141 | 26.12 | |
| Mean ± SD | 1.57 ± 1.23 | | 1.50 ± 1.07 | | 0.339 |
| **Gender** | | | | | 0.583 |
| Boy | 2,951 | 59.34 | 11,595 | 58.91 | |
| Girl | 2,022 | 40.66 | 8,088 | 41.09 | |

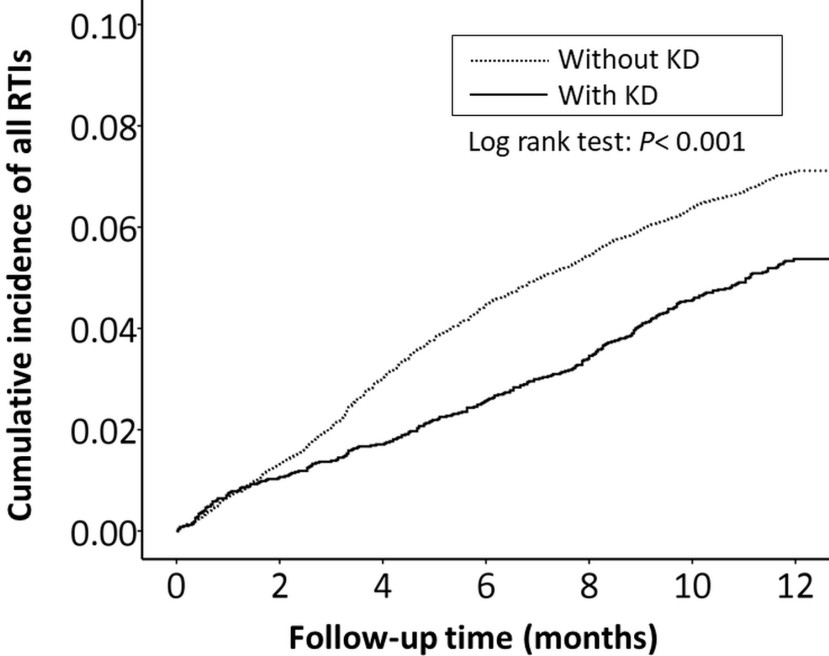

**Figure 2  The Kaplan–Meier curve showed the accumulative incidences of all respiratory tract infection-related hospitalizations between KD cohort and control cohort by time ($p < 0.001$).**

for almost all RTI-related hospitalizations (aHR: 0.75; 95% CI [0.66–0.85]; $p < 0.001$), pneumonia-related hospitalizations (aHR: 0.64; 95% CI [0.52–0.79]; $P < 0.001$), AOM-related hospitalizations (aHR: 0.61; 95% CI [0.42–0.90]; $p < 0.05$), acute bronchiolitis-related hospitalizations (aHR: 0.77; 95% CI [0.560–0.99]; $p = 0.042$), and tonsillitis (aHR: 1.04; 95% CI [0.76–1.41]; $p = 0.845$). The reduction of RTI-related hospitalizations was observed in both girls and boys. Girls in the KD cohort displayed a 0.75-fold decreased incidence of RTI-related hospitalizations compared with girls in the control cohort (girls: aHR, 0.75; 95% CI [0.60–0.93]; $P < 0.05$; boys: aHR, 0.74; 95% CI [0.63–087]; $p < 0.001$). Moreover, the incidence of RTI-related hospitalizations increased with age in the KD cohort whereas decreased with age in the non-KD cohort. The highest incidence of RTI-related

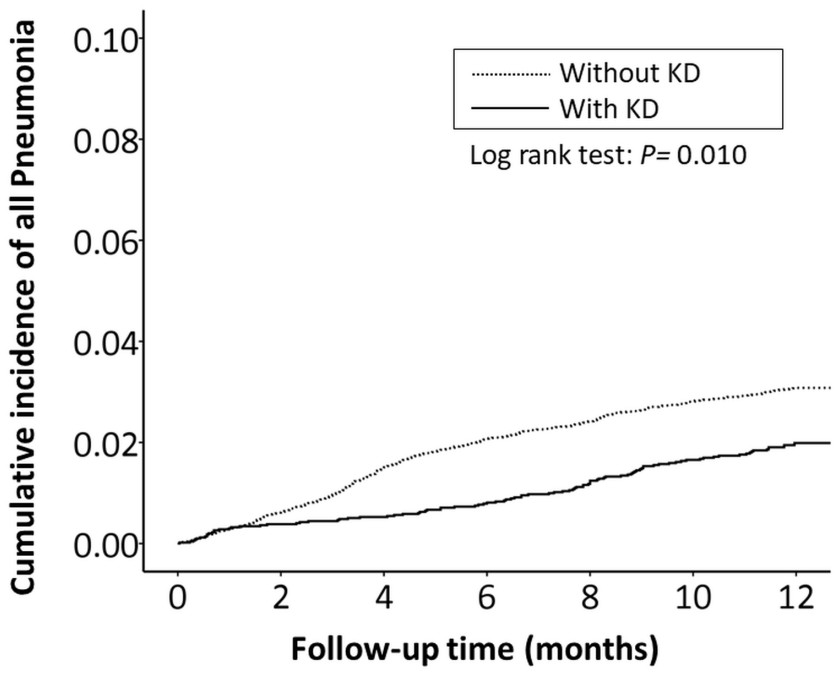

**Figure 3** The Kaplan–Meier curve showed the accumulative incidences of pneumonia-related hospitalizations between KD cohort and control cohort by time ($p = 0.01$).

hospitalizations was seen in non-KD group patients aged 0–2 years (19.34 per 1,000 person-months). The risk for RTI-related hospitalizations was lower in the KD group than in the non-KD cohort for patients aged 0–2 years: aHR, 0.69; 95% CI [0.59–0.80]; $P < 0.001$).

We further analyzed the aHR of RTI-related hospitalizations within the 12-month follow-up period to investigate the affecting duration of IVIG. We further stratified the episodes of hospitalizations by follow-up time into four periods (Table 3). The incidence of RTI-related hospitalizations in both cohorts decreased as the follow-up time increased in the first six months. In these follow-up periods, the risk of RTI-related hospitalizations was lower in the KD cohort than in the control cohort and the lowest aHR was observed in the three-to-six month follow-up group (aHR: 0.49; 95% CI [0.37–0.64]; $p < 0.001$).

## DISCUSSION

Research evidence in the area of susceptibility to infection in patients with KD is scarce and we found a decreased risk of subsequent hospitalizations due to pneumonia, AOM, and acute bronchiolitis in patients with KD within 6 months after discharge. We procured a marked lower aHR of 0.75 (95% CI [0.66–0.85]) for RTI-related hospitalizations in the KD cohort than in the non-KD cohort. The protective effects persisted during the 6-month follow-up period with the lowest aHR within the 3–6 months (aHR: 0.49, 95% CI [0.37–0.64]). Some protective effects from RTI-related hospitalizations in the KD cohort

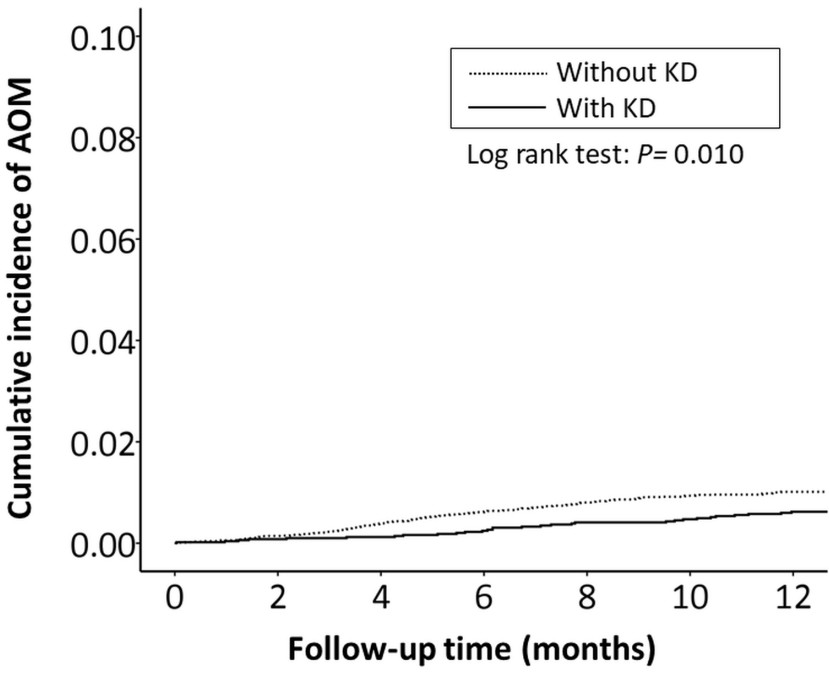

**Figure 4** The Kaplan–Meier curve showed the accumulative incidences of AOM-related hospitalizations between KD cohort and control cohort by time ($p = 0.01$).

seem to exist compared with the non-KD cohort. Peak protective effects were within the first 3–6 months.

Although it has been frequently postulated that KD is caused by an aberrant immune response after an infectious episode, the pathogenesis of KD is not fully understood. KD patients are not regarded as immunocompromised individuals, alterations of immune systems are believed to play important roles in KD (*Hara et al., 2016*; *Newburger, Takahashi & Burns, 2016*; *Wang et al., 2005*). In the acute stage of KD, evidence has shown that the roles of T cell activation and inflammatory cytokines are both critical (*Brogan et al., 2008*; *Lee et al., 2015*). T cells such as Th1, Th2, and Treg cells have all been identified to be involved in this stage. Cytokines derived from Th1 cells (IL-2, IFN-$\gamma$, and IL-10) and Th2 cells (Il-4 and IL-5) have also been identified to be involved in the disease process (*Abe et al., 2005*; *Hsieh et al., 2011*; *Kimura et al., 2004*; *Matsubara, Ichiyama & Furukawa, 2005*). T cell activation, along with cytokine-induced macrophage activation, is critical to the pathogenesis of vascular endothelial damage. The subsequent infiltration of neutrophils, plasma cells, and eosinophils in coronary arteries may cause the destruction of arterial wall integrity and result in dilatation and aneurysm formation. Treg cells play an important role in weakening the pathogenic effects of T cells in the destruction of coronary arteries (*Ye et al., 2016*). Further studies have demonstrated significantly lower levels of Treg-related gene expressions such as FOXP3, GITR, and CTLA4 in acute KD patients prior to treatment compared with healthy controls (*Anthony & Ravetch, 2010*; *Ephrem et al., 2008*). On the other hand, the mechanism of immune regulation by IVIG in KD patients remains unclear

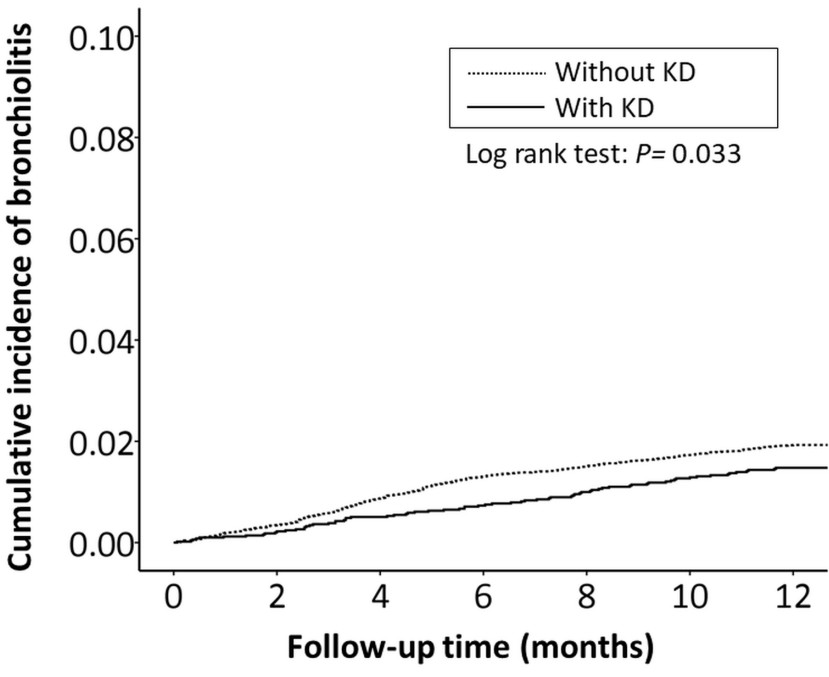

**Figure 5** The Kaplan–Meier curve showed the accumulative incidences of bronchiolitis-related hospitalizations between KD cohort and control cohort by time ($p = 0.033$).

(*Burns & Franco, 2015*; *Ephrem et al., 2005*; *Perez et al., 2017*; *Siberil et al., 2007*). Many potential mechanisms have been proposed, such as agent-specific neutralizing antibodies, decreased proliferation of Th17 cells, and reduced cytokine release. IVIG stimulates an immature myeloid population of IL-10-secreted DCs, which leads to the expansion of induced Treg cells. Treg cells then recognize the Fc of IgG and block the activated Fcγ receptor and stimulate the inhibitory Fcγ RIIb receptor. In addition, IVIG contains various antibodies specific to a vast range of pathogens and it may be taken for granted that IVIG protects the KD group from subsequent RTI-related hospitalizations. However, immunoinflammatory responses are complex and both KD and IVIG may contribute to the reduction of RTI-related hospitalizations. Further studies are warranted to clarify the entire mechanisms of the observed protective effects in present study.

Prophylactic IVIG use in immunocompromised patients has been well documented but studies investigating IVIG use in non-immunocompromised patients are limited (*Keller & Stiehm, 2000*; *Mouthon & Lortholary, 2003*; *Orange et al., 2010*; *Perez et al., 2017*). We found an obvious reduction of risk for RTI-related hospitalizations in IVIG-treated patients. However, in addition to flu-like symptoms, severe adverse effects may occur after IVIG use, such as anaphylaxis and thrombo-embolism (*Hefer & Jaloudi, 2004*; *Milani, Dalia & Colvin, 2009*). Regularly prophylactic IVIG use in immunocompetent patients is not suggested. For high risk groups, costs and benefits should be evaluated carefully to make a decision of IVIG use (*Ohlsson & Lacy, 2013*; *Perez et al., 2017*). In the present study, no severe adverse effects were noted after IVIG use.

**Table 2  Incidence and aHR of respiratory tract infection-related hospitalization stratified by sex, age, between KD and non-KD cohorts.**

| Variables | KD | | | Non-KD | | | Compared with non-KD | |
|---|---|---|---|---|---|---|---|---|
| | Event | Person months | Rate | Event | Person months | Rate | IRR (95% CI) | aHR (95% CI) |
| **Overall infection rate** | | | | | | | | |
| All respiratory | 267 | 1,597 | 16.72 | 1,400 | 7,298 | 19.18 | 0.87 (0.76–0.99)[*] | 0.75 (0.66–0.85)[**] |
| Pneumonia | 97 | 614 | 15.79 | 595 | 2,899 | 20.52 | 0.77 (0.61–0.96)[*] | 0.64 (0.52–0.79)[**] |
| AOM | 30 | 201 | 14.93 | 193 | 1,021 | 18.89 | 0.79 (0.52–1.17) | 0.61 (0.42–0.90)[*] |
| Bronchiolitis | 72 | 424 | 16.99 | 370 | 1,840 | 20.10 | 0.85 (0.65–1.09) | 0.77 (0.60–0.99)[*] |
| Tonsillitis | 51 | 294 | 17.35 | 192 | 1,160 | 16.55 | 1.05 (0.75–1.43) | 1.03 (0.76–1.40) |
| **Sex** | | | | | | | | |
| Boy | 176 | 1,060 | 16.60 | 917 | 4,723 | 19.41 | 0.85 (0.72–1.01) | 0.75 (0.64–0.88)[**] |
| Girl | 91 | 536 | 16.97 | 483 | 2,574 | 18.76 | 0.90 (0.71–1.13) | 0.75 (0.60–0.94)[*] |
| **Age (y)** | | | | | | | | |
| 0–2 | 209 | 1,296 | 16.13 | 1,185 | 6,127 | 19.34 | 0.83 (0.72–0.97)[*] | 0.69 (0.59–0.80)[**] |
| 2–6 | 58 | 301 | 19.25 | 215 | 1,171 | 18.36 | 1.05 (0.77–1.41) | 1.07 (0.80–1.43) |

**Notes.**
  [*] $p < 0.05$.
  [**] $p < 0.001$.
  IRR, incidence rate ratio; aHR, multiple analysis including sex, age; Rate, incidence rate (per 1,000 person months).

**Table 3  Incidence and aHR of respiratory tract infection-related admission between KD and non-KD cohorts within the one-year follow up.**

| Variables | KD | | | Non-KD | | | Compared with non-KD | |
|---|---|---|---|---|---|---|---|---|
| | Event | Person months | Rate | Event | Person months | Rate | IRR (95% CI) | aHR (95% CI) |
| **Follow-up time, (m)** | | | | | | | | |
| 0–3 | 69 | 83 | 83.50 | 405 | 636 | 80.64 | 1.31 (1.00–1.70)[*] | 0.68 (0.53–0.88)[*] |
| 3–6 | 59 | 268 | 21.98 | 475 | 2,079 | 22.65 | 0.96 (0.72–1.26) | 0.49 (0.37–0.64)[**] |
| 6–9 | 74 | 568 | 13.02 | 290 | 2,175 | 13.34 | 0.98 (0.75–1.26) | 0.99 (0.77–1.28) |
| 9–12 | 65 | 677 | 9.60 | 230 | 2,408 | 9.55 | 1.00 (0.75–1.33) | 1.10 (0.84–1.45) |

**Notes.**
  [*] $p < 0.05$.
  [**] $p < 0.001$.
  IRR, incidence rate ratio; aHR, multiple analysis including sex, age; Rate, incidence rate (per 1,000 person months).

For patients with PID, monthly IVIG supplement is suggested (*Hemming, 2001*; *Keller & Stiehm, 2000*; *Wong & White, 2016*). Waning of immunoglobulin is a main concern of IVIG treatment. Half-lives of immunoglobulins vary from 2.5 to 23 days in previous reports (*Bonilla, 2008*; *Leuridan et al., 2010*). The required serum IgG level for adequate protection and the achieved concentration after IVIG supplement are different in different IVIG products and dosages (*Orange et al., 2010*; *Perez et al., 2017*). Our study showed the protective effects could persist up to six months after IVIG administration, with a peak in 3–6 months. For patients with KD, high dose of IVIG treatment (2 g/kg/dose) is the standard treatment and higher dose may contribute to longer protection. Furthermore, we are curious how long will the protective effects exist. Within the current 1-year follow-up period, the aHR of RTI-related hospitalizations decreased in the KD group in 0–3 months and 3–6 months (Table 3, aHR: 0.68 and 0.49, respectively) whereas the aHR of RTI-related

hospitalization declared no difference between the KD and non-KD group in 6–9 and 9–12 months. The abrupt disappear of decreasing aHR for RTI-related hospitalization 6 months after KD may suggest that waning of IVIG-containing antibodies occurred. It also provides a hint the influences of IVIG exist; differences in infectious susceptibility of KD patients may not change by time. Moreover, it seems that patients in KD cohort were less susceptible to pneumonia, AOM, and bronchiolitis but not tonsillitis. Therefore, IVIG may have different protective effect on different type of respiratory tract infectious disease. Longer follow-up periods may tell us the exact duration of protection on different respiratory tract infectious disease and the baseline infectious susceptibility in both groups. Further studies investigating the optimal dosage and interval of IVIG are also required.

The strength of current study is a large population with standardized treatment. There were several limitations to this study. First, the observed protective effects may attribute to different susceptibility to infection in patients with KD itself or the immune regulatory effects of IVIG. It's valuable to compare the infection risk in KD patients without IVIG but IVIG is the standard treatment of KD. It's unethical to not use IVIG in patients with KD. Similarly, it's more meaningful to compare the incidences of infection in general populations with and without IVIG. However, IVIG is not commonly used in general population and it's difficult to recruit healthy individuals to receive IVIG. Further studies are warranted to clarify the possible role of IVIG. Second, although one-year follow-up period is not short, longer follow-up time will provide more information regarding the duration of protection. Third, RTI-infected patients treated as outpatients with oral antibiotics were not included in our analysis. Although NHI in Taiwan covered nearly all the populations and paid most of the hospitalization fees, some parents chose to treat their children's illness such as pneumonia, AOM, and acute bronchiolitis at home. Thus, the incidence of RTI-related hospitalization may have been underestimated for both groups. The observed lower risk may indicate the less severe infection in patients with KD. Furthermore, the NHIRD does not contain complete information such as laboratory data and image reports. Hence, the causative pathogens of the RTIs remained unknown. The severity of KD were not analyzed. Moreover, although the standard treatment of KD is IVIG administration as 2g/kg/dose in Taiwan, individualized treatment may occur and a small part of KD patient may receive different dosage of IVIG or no IVIG treatment. (*Lin et al., 2015*). The details of IVIG use were not available and subgroup analyses were not performed. Finally, the current retrospective cohort study could not clarify the underlying mechanism of the protective effects from RTI-related hospitalizations in KD patients with IVIG treatment. Further investigation of these factors with a well-designed prospective cohort study with close monitoring of KD patient serum markers in the convalescent stage is warranted.

## CONCLUSIONS

In conclusion, this study found that children with KD had an approximately quarter decreased risk of RTI-related hospitalization in the subsequent one year after IVIG treatment, regardless of age, sex. This protective effect is not well understood but seemed

to be well maintained for at least for six months and then gradually decreased. Further researches are warranted to illustrate the underpinning mechanisms and clarify the possible role of IVIG.

## ACKNOWLEDGEMENTS

The authors would like to thank Dr. Hou-Ling Lung for providing the statistical opinion.

### Funding

The authors received no funding for this work.

### Competing Interests

The authors declare there are no competing interests.

### Author Contributions

- Wei-Te Lei and Chien-Yu Lin conceived and designed the experiments, performed the experiments, analyzed the data, authored or reviewed drafts of the paper, approved the final draft.
- Yu-Hsuan Kao and Chao-Hsu Lin performed the experiments, prepared figures and/or tables, authored or reviewed drafts of the paper, approved the final draft.
- Cheng-Hung Lee performed the experiments, contributed reagents/materials/analysis tools, prepared figures and/or tables, authored or reviewed drafts of the paper, approved the final draft.
- Shyh-Dar Shyur and Kuender-Der Yang performed the experiments, authored or reviewed drafts of the paper, approved the final draft.
- Jian-Han Chen conceived and designed the experiments, performed the experiments, analyzed the data, contributed reagents/materials/analysis tools, authored or reviewed drafts of the paper, approved the final draft.

### Human Ethics

The following information was supplied relating to ethical approvals (i.e., approving body and any reference numbers):

This study was reviewed and approved by the Institutional Review Board of Buddhist Tzu Chi Hospital, Dalin, Taiwan (IRB approval number: B10503021).

### Data Availability

The raw data is provided as a Supplemental Information 1.

### Supplemental Information

Supplemental information for this article can be found online at http://dx.doi.org/10.7717/peerj.4539#supplemental-information.

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
