# Peer review of "The risk of hospitalization for respiratory tract infection (RTI) in children who are treated with high-dose IVIG in Kawasaki Disease: a nationwide population-based matched cohort study"

_PeerJ, doi:10.7717/peerj.4539_

## Round 0.1 · original submission · Major Revisions

Dear Author,
The manuscript highlighted some interesting points spotted by reviewers, however there are relevant concerns to be addressed. Hope you will be able to make necessary amendments before we can further conside your submission

Kind regards

Reviewer 1 ·

Basic reporting

The title of the study does not adequately reflect the contents of this study. This is a follow-up study for children who are treated with high-dose IVIG in KD. It is not appropriate to use the observation results as a title. What about this title: ‘the risk of hospitalization for respiratory tract infection (RTI) in children who are treat with high-dose IVIG in KD.’

In addition, it is necessary to describe constantly the names of cohort groups in all text and the tables. There are various expressions in the text and the tables, which can confuse readers. For example, it is expressed as ‘KD group’ or ‘IVIG group’ or ‘Patient without KD’ or ‘non-KD’, and so on.

Experimental design

There is a concern about the experimental design.

First of all, comparing the comorbidity (preterm, heart disease, asthma, allergic rhinitis) of both groups, statistically significant differences are shown. Are these control groups unsuitable for matched cohort groups? In my opinion, that is considered to be the most influential factor in the hospitalization of RTI. Therefore, I would like to ask you to select the control group again or to use another statistical methods (for example, instrumental variable analysis, …) to adjust the bias from confounding factors.

Second, why did you choose pneumonia, AOM, and bronchiolitis for RTI in this study. It is not clear whether the selected RTI is caused by a frequency of illness, or whether it is based on a common pathogen or anatomical position of respiratory tract. Please add a description for this.

Validity of the findings

There is a mistake in statistics. Eighteen children have congenital heart disease in KD group, which was calculated as 0.35%. It is 0.13% that I calculate. Thus, the expressions of 186 lines and table 1 should be corrected. It also affects discussion, so it needs modification

Additional comments

I think it is an honor to review the paper on interesting research. So far, numerous studies have been conducted regarding the appropriate treatment methods and outcomes, including the pathophysiology in Kawasaki disease (KD). However, except for the prevalence of allergic diseases, there are few observational follow-up studies for patients hospitalized with KD. This study is considered to be very interesting attempt to investigate the incidence of hospitalization by the respiratory tract infection (RTI) in children who are treated with high-dose IVIG in KD. Particularly, I appreciate the authors ' efforts to explore the vast amounts of data which have been provided by NHIRD.

Here are some of the following comments as a reviewer to make a more meaningful study.

Thank you!

Reviewer 2 ·

Basic reporting

1. Table 1: The group of 13709 persons should be "non-KD group".

Experimental design

1. The patients of Kawasaki disease was recruited by ICD codes only. However, there is a certain part of patients who were admitted with the main diagnosis of Kawasaki disease did not receive IVIG therapy in Taiwan. (J Chin Med Assoc. 2015 Feb;78(2):121-6.) If the patients can be further grouped according to receive IVIG or not, the causal relationship between the decreased risk of hospitalization and IVIG would be further strengthened.

Validity of the findings

1. From methodology point of view, the authors used age, gender, preterm, heart disease, asthma, allergic rhinitis, and pre-RTIs to build propensity scores. However, they did not describe the detail of the propensity scores, such as the matching method, c statistics, and propensity scores distributions.

2. I would like to suggest the authors to select the control group (1:4 or 1:10) randomly then adjust age, gender, preterm, heart disease, asthma, allergic rhinitis, and pre-RTIs by traditional logistic models to see if there is different results.

Additional comments

This article is well structured and written in good English.

---

## Round 0.2 · Major Revisions

Although the manuscript has been improved, there are key points that need to be addressed (particular attention to reviewer 2 comments.

Reviewer 1 ·

Basic reporting

I think the paper was properly revised and supplemented.

Experimental design

I think the paper was properly revised and supplemented.

Validity of the findings

Additional modifications are recommended as follows:
First, the spell ‘s’ is missing from Table 3 (person months), and must be corrected.
Second, how about you modifying the expression in line 285 as follows?
from “… but seemed to persist at least for 6 months.”
to “… but seemed to be well maintained for at least 6 months and then gradually decreased.

Additional comments

I think the paper was properly revised and supplemented.

As researchers have mentioned, further study into the effectiveness of IVIG’s protection against infectious disease is thought to be interesting; according to the dosages of IVIG, according to the pathogen type (viral or bacterial), and so on.

Thank you for your every efforts!

Reviewer 2 ·

Basic reporting

I have no further comments.

Experimental design

IVIG therapy has strong impact on the subsequent respiratory tract infection. I don’t think it can be ignored in this study. Furthermore, as the authors wrote, some children diagnosed with KD may present with concomitant respiratory tract infections and may pose a higher risk for subsequent RTI-related hospitalizations.

Validity of the findings

I don’t think building propensity score from two cofactors can reduce bias comparing with 7 factors’ model in the previous manuscript. Furthermore, urbanization, allergic disease, and congenital diseases also have strong impact on the subsequent RTI. I didn’t see any effort that the authors trying to control them.
I still suggest that randomly selecting patients of control group matched by age, gender, allergic disease, congenital disease, and urbanization.

Additional comments

This article in interesting. However, the authors should revise the manuscript before it can be accepted.

---

## Round 0.3 · accepted · Accept

The manuscript has thoroughly fulfilled the reviewers comments.